# The Enduring Challenge of Literacy Issues in Adulthood: Investigating Spelling Deficits among Dyslexic Italian University Students

**DOI:** 10.3390/brainsci14070712

**Published:** 2024-07-15

**Authors:** Francesca Vizzi, Marika Iaia, Maria Diletta Carlino, Chiara Valeria Marinelli, Marco Turi, Paola Angelelli

**Affiliations:** 1Department of Human and Social Sciences, University of Salento, 73100 Lecce, Italy; francesca.vizzi@unisalento.it (F.V.); marika.iaia@unisalento.it (M.I.); mariadiletta.carlino@unisalento.it (M.D.C.); marco.turi@unisalento.it (M.T.); 2Laboratory of Applied Psychology and Intervention, Department of Experimental Medicine, University of Salento, 73110 Lecce, Italy; 3Cognitive and Affective Neuroscience Laboratory, Department of Humanities, University of Foggia, 71122 Foggia, Italy; chiaravaleria.marinelli@unifg.it

**Keywords:** spelling, spelling errors, dyslexia, adults, transparent orthography

## Abstract

The issue of literacy challenges among dyslexic adults remains a significant concern. This study investigates spelling deficits among highly educated adults with dyslexia learning a transparent orthography. Thirty-eight Italian dyslexic university students were examined and compared to a group of age- and education-matched typical readers. Firstly, we analyzed spelling performance using a Passage Dictation Test. Additionally, lists of words varying in length and word frequency were dictated under two experimental conditions: a normal condition (NC) and an articulatory suppression condition (ASC). The ASC assessed the participants’ ability to spell with interference to the phonological (sublexical) spelling procedure, i.e., the most likely compensated spelling strategy of Italian dyslexic spellers. The results clearly indicated that, in spelling the meaningful passage, dyslexic participants underperformed compared to the controls, with a prevalence of lexical errors, despite the comparison with the normative reference data showing only mild spelling difficulties. In spelling isolated words in normal conditions, dyslexic participants performed within the reference norms and as accurately as control participants across all stimuli (short words, high- and low-frequency words), except for long words, where their spelling difficulties were evident. Articulatory suppression significantly impaired dyslexics’ performance on short stimuli, reducing the usual sublexical advantage associated with them, and exacerbated misspellings on long words. Additionally, articulatory suppression disproportionately affected dyslexics’ performance on high-frequency words, diminishing the typical lexical advantage associated with these words. Results are discussed in terms of their theoretical, clinical, and educational implications.

## 1. Introduction

Literacy acquisition deficits are neurodevelopmental disorders that persist throughout life [1,2,3,4,5]. Although in recent years the manifestations of reading and spelling problems in adulthood have garnered increased attention, they remain an underexplored area compared to the large number of studies focusing on developmental age [6,7,8]. Follow-up studies indicate that children with dyslexia continue to experience reading and spelling difficulties [9,10]. However, the manifestation of these disorders varies based on their severity, as well as individuals’ cognitive resources, educational opportunities [11,12,13], and the consistency of the orthography in which they learn [14,15,16,17]. Literacy acquisition deficits can lead to underachievement through the school years and beyond, resulting in negative consequences in academic and professional settings. This can potentially lead to frustration, as well as low levels of self-esteem and motivation [1,18,19,20,21,22]. Additionally, experiencing challenges in education can result in unemployment, reduced job satisfaction (including lower wages and fewer promotions, as indicated by Witty et al. [23]), and subsequent issues of a psychological, economic, and social nature [24,25]. Gaining a thorough understanding of the features of developmental literacy deficits in adults is essential for improving the sensitivity of diagnostics and the appropriateness of the intervention.

Detecting the characteristics of developmental literacy problems in adults can be challenging for several reasons. Some studies, in fact, highlight compensatory effects associated with advancing age and schooling, likely due to increased exposure to education. For example, studies comparing adults and children with dyslexia have found that, despite adults suffering from persistent difficulties in pseudoword reading, they were more efficient (increased speed and accuracy) than children in word recognition in lexical decision tasks (e.g., [26]). According to the authors, over years of exposure to written language, adults with dyslexia build a database of orthographic word representations, enabling more efficient recognition. Additionally, in meaningfully connected texts, they demonstrated improved accuracy in word decoding [26], whereas they struggled more with nonsense-passage reading [27]. Indeed, for adults with dyslexia, context acts as another compensatory tool, streamlining reading.

Fewer studies focus on spelling deficits with respect to reading impairment among adults with reading disorders. However, spelling is the most prominent marker of developmental literacy problems in adults [28,29,30,31]. Moreover, there is ample evidence that spelling difficulties not only persist, but are also more severe [3,16,32,33,34,35]. In fact, spelling disorders could be one of the most frequent challenges among adults with dyslexia [16,36,37]. Specifically, dyslexic adults write more slowly than normal readers of a similar age and education, and their written productions are poorer with more spelling errors [16,18,38,39,40].

Evidence on the main characteristics of spelling problems in adults is primarily from studies in opaque orthographies like English [41,42]. Languages vary in phonology–orthography consistency [43]. For example, Italian has predictable spelling-to-sound correspondence, unlike English or French [44,45]. A study [46] analyzed orthographic transparency in seven languages using entropy values to measure letter-to-phoneme ambiguity. English had the highest ambiguity (>0.5), while Italian and Hungarian were the most predictable (0.2 and 0.1, respectively). Differences in the degree of consistency between orthographies affect spelling and reading acquisition. Cross-linguistic studies carried over mainly in a developmental context found lower spelling performance in inconsistent orthographies (e.g., English) compared to children learning more consistent ones (for Czech: [47,48]; for Italian: [49]). Moreover, these studies show that different cognitive competencies and spelling procedures are employed in languages with different degrees of consistency [43]. Indeed, according to dual-route models (e.g., [50,51,52,53]), spelling depends on the efficiency of both phonological and lexical processes, as well as their interaction. The lexical procedure retrieves spellings of known words from the orthographic lexicon (it is the only procedure that allows the transcription of irregular or unpredictable stimuli), while the sublexical procedure constructs spellings for unfamiliar words and nonwords using phonological-to-orthographic rules. Failures in the lexical procedure result in phonologically plausible errors, where words are phonetically correct but spelled incorrectly (e.g., the Italian word “kwota”, (rate), is spelled as “CUOTA*” instead of “QUOTA”). These misspellings may be due to overreliance on the sublexical (phonological) spelling procedure. The sublexical procedure involves two conversion processes: acoustic-to-phonological conversion, which segments and identifies the phonological string, and phoneme-to-grapheme conversion, which applies rules for sound-to-spelling mappings [52]. Sublexical inefficiencies may lead to phonological errors that alter the phonemic structure of stimuli (e.g., substitutions, inversions, insertions, and omissions of letters, for example, “CUODA*” instead of “QUOTA”). Phonological errors may reflect difficulties in phonemic segmentation and specification, as well as phoneme-to-grapheme encoding and phonemic and graphemic buffer disorders. Both procedures can function independently and can be individually impaired [54,55], but evidence shows they interact in typical and dyslexic spellers ([56,57]; for Italian see [58,59,60]).

Cross-linguistic studies clearly indicate differences in the reliance on different spelling procedures as a function of the degree of orthographic consistency: the less regular the writing system, the less children rely on sublexical processing. This pattern has been reported comparing English children to Spanish, German, and French children (for reviews, see [47,61]). In other words, while English children rely more on the lexical procedure, in more consistent orthography, there is an overreliance on the sublexical procedure (for Italian, see [49]). The sublexical spelling procedure is acquired rapidly in more consistent orthographies (for Italian, see [62]; for Czech, see [48]; for German, see [63]) after only a few years of schooling, while acquiring spelling skills based on lexical knowledge is slower because it requires the acquisition of each word as an individual item. In the study by Notarnicola et al. [62], typical Italian readers optimized their spelling of regular words within the first three years of schooling. However, they did not fully master the spelling of words with unpredictable spellings, which requires the retrieval of lexical orthographic representations, even by the end of middle school.

The relatively easy optimization in the use of the sublexical spelling procedure was demonstrated even in individuals with dyslexia. Dyslexic children may overcome their difficulties with the sublexical spelling procedure more easily than with the lexical one (for German, see [63,64]; for Italian, see [65]). Regarding German, Wimmer [63] retrospectively examined the early difficulties of 12 Austrian dyslexic children, diagnosed after four years of schooling. He evaluated their spelling performance with words and nonwords, analyzing error types. In first grade, the children could not spell a single nonword, with many errors being phonological. By fourth grade, spelling deficits persisted but were more selective, primarily affecting tasks requiring lexical knowledge. Only 2% of words still had phonological errors. Their nonword spelling was consistently nearly perfect. Overall, these data indicate a gradual but slow improvement in phonological encoding in spelling, alongside persistent lexical deficits. Similarly, in Italian, a study by Angelelli et al. [65] compared third- and fifth-grade dyslexic children’s spelling performance. Third-graders performed worse than controls across all stimuli, making both a high rate of phonologically plausible and phonological errors. Fifth-graders performed well on phoneme-to-grapheme transcription stimuli (words and nonwords), but struggled with unpredictable transcription words. Compared to controls, dyslexics showed significantly more phonologically plausible errors, with phonological ones decreasing considerably, resulting in similarities between groups.

Overall, it was hypothesized that, with the support of a consistent orthography and a phonics teaching program, spellers suffering from learning disorders will persevere in phonological encoding thanks to the better optimization of this procedure. Due to the high consistence of transparent orthographies, a discrete level of accuracy can be achieved [63,65]. In other words, in transparent orthographies, the improvement in the ability to spell using phoneme-to-grapheme conversion procedures serves as a compensation tool for individuals with dyslexia, who continue to experience lexical spelling deficits, i.e., a deficit in memorizing and retrieving the orthographic representation of words with unpredictable spelling (for a longitudinal study on Italian children with dyslexia, see also Marinelli, Cellini, Zoccolotti, and Angelelli [60]). However, cases of significant difficulties in the sublexical spelling procedure were also reported in children learning a transparent orthography [66]. In the study, the spelling performance of Italian dyslexic children with and without a history of oral language delay in preschool years was analyzed. The results showed that, among dyslexic children, those who suffered from language delay were generally more compromised, and their errors were of a phonological nature (errors in transcribing stimuli containing geminate consonants, non-continuant consonants, and polysyllabic stimuli).

However, despite the need for caution in generalizing findings from opaque to consistent orthographies and from dyslexic children to dyslexic adults (due to age-related changes in phenomenology and different reliance on compensatory mechanisms), studies on the spelling skills of dyslexic adults learning a transparent orthography are notably scarce. Afonso and colleagues [18] compared Spanish adults with dyslexia to normal age-matched readers and revealed that the former were significantly slower and committed more errors than the controls in spelling-to-dictation and direct copy transcoding tasks. They showed impaired access to orthographic representations, confirming the prevalent reliance on sublexical spelling.

Regarding the Italian language, according to our knowledge, very little is known about literacy deficits in adults. A longitudinal study [67] following 33 dyslexic adults, previously diagnosed when they were children, confirmed the persistence of deficient reading skills despite a relative improvement in reading speed, and long-lasting deficits in metaphonological abilities. Unfortunately, the study did not report the spelling outcome. Re et al. [8] conducted a study comparing the reading and spelling abilities of 24 Italian university students with dyslexia and 99 adults. They found significant differences in reading fluency and accuracy across all reading tasks between individuals with dyslexia and the control group. Moreover, dyslexic individuals demonstrated poorer spelling performance compared to the controls. In particular, the authors used a spelling-to-dictation task of single words of three or four syllables in two experimental conditions: normal and with an articulatory suppression condition. While there was no significant difference between the groups under normal conditions (with both groups generally performing well), a notable difference emerged under the suppression condition: the error rate for individuals with dyslexia was six times higher compared to the control group. According to the authors, articulatory suppression hinders the use of the articulatory loop, which is crucial for less experienced writers who likely depend on underlying articulatory processes.

Overall, in the present study, we aim to provide further evidence on the presence and characteristics of spelling deficits in highly educated Italian adults with dyslexia and highlight the possible loci of fragilities. Firstly, we analyzed spelling performance using a Passage Dictation Test, a more organic task in which diverse error types may occur, rather than only single-word spelling errors (for example, lexical spelling inefficiencies can manifest in the failure to respect word units, with erroneous blending or separated words; see [49]). Moreover, the test allows us to analyze the persistence of spelling difficulties despite the support provided by contextual information in a meaningful passage. Additionally, we dictated single words varying in length (a source of phonetic–phonological complexity in the sublexical spelling procedure) and frequency (a variable modulating the lexical spelling procedure). The length of the stimulus is a well-documented variable that complicates the sublexical spelling procedure [68,69,70]. Conversely, word frequency is a well-known lexical factor that supports lexical activation in spelling, particularly for high-frequency words, resulting in higher accuracy. Word frequency effects in spelling have been observed in both adults and children [60,71,72], with higher accuracy for high-frequency words compared to low-frequency ones. According to dual-route models [73], the activation rate via the lexical pathway is proportional to word frequency, allowing more frequent words to be processed more quickly. The words were dictated in a normal condition (NC) and an articulatory suppression condition (ASC).

The experimental paradigm of articulatory suppression is particularly intriguing, not only due to its dual-task nature and attentional demands, but especially because it interferes with phonological processes. Colombo et al. [74] conducted a study to explore the impact of concurrent articulatory suppression versus a non-phonological interfering task (foot tapping) on the spelling of normal adults. If no difference emerged between the two conditions causing interference, it would suggest that effects are situated at the level of attention, stemming from fewer resources being accessible to the executive system. Moreover, they investigated the effects of concurrent articulatory suppression on spelling in two conditions: words presented orally and visually. The results showed more spelling errors with oral presentation than visual, and poorer performance under articulatory suppression compared to foot tapping or no interference. Concurrent articulatory suppression, which engages the short-term memory of the articulatory loop, affects the phonemic segmentation, storage, and maintenance of resulting phonological specifications in the phonemic buffer (for a review, see [75]; also see [76] and [77]). These processes constitute the initial computations of the sublexical spelling strategy. Additionally, its attentional demands may interfere more heavily with sequential operations, which are also characteristic of the sublexical strategy.

Interfering with the sublexical spelling procedure and evaluating its level of automatization may be particularly interesting, considering that in transparent orthographies, many words can be spelled sublexically (see [75]); misspellings in highly educated individuals may be few, and sublexical spelling is the primary developmental compensatory strategy for individuals with dyslexia [65]. Moreover, this paradigm may be particularly suited as it could worsen the lexical deficit by impeding compensation through the sublexical procedure. Interestingly, in the study on acquired spelling disorders by Folk and Jones [77], articulatory suppression was employed to isolate the already weakened lexical spelling system in the two patients studied by disrupting the contribution of the sublexical spelling procedure.

## 2. Method

### 2.1. Participants

The participants were 38 Italian university students (15 M, 23 F) with developmental dyslexia with a mean age of 22.2 years (SD = 3.07). The selection criteria for inclusion in the dyslexic group were as follows: (i) reading delay on a standard passage reading test and/or a single word and nonword reading test (at least 1.65 SDs below the mean of the normative sample for reading speed or accuracy in the LSC-SUA reading tests [78]; for the description, see Section 2.2.2); (ii) and intelligence level in the normal range (i.e., Full Scale Intelligence Quotient (FSIQ) ≥ 85, as measured by the Wechsler Intelligence Scale for Adults—IV; normative Italian data by Orsini and Pezzuti [79]; see Section 2.2.1). Among the dyslexic participants, 18 showed impairments in both accuracy and speed, while 16 exhibited impairments only in accuracy and 4 only in speed. The criteria for inclusion considered selective impairment in either speed or accuracy, as research has indicated that dyslexic individuals may adapt their reading abilities strategically, either prioritizing speed (at the expense of accuracy) or accuracy (at the cost of speed) ([80]; see also [81]).

Twenty-seven university students (8 M, 19 F), with a mean age of 22.8 years (SD = 2.43), participated in the experiment as control subjects. All control subjects performed within the norm for both reading speed and accuracy in both reading tests and had an intelligence level in the normal range.

All study participants were assessed for their cognitive, reading, and spelling abilities at the time of enrollment by our psychodiagnostic service at the University of Salento. Therefore, the profiles we describe in the study are current. However, thirteen of the dyslexic participants had a previous diagnosis of dyslexia (generally dating back to high school years), but they were re-evaluated at the time of enrollment in the study. An accurate collection of anamnestic information, including school history, family history of learning disorders, and perceived difficulties in daily life literacy activities, was conducted to corroborate (or exclude in the case of control participants) the psychometric diagnosis of dyslexia.

The demographic and clinical data of both groups are presented in Table 1.

The two groups were similar in age (t_(63)_ = 0.85; n.s.) and gender distribution (X^2^ = 0.66; n.s.). The Full-Scale Intelligence Quotient (FSIQ) was slightly lower in the dyslexic group (t_(63)_ = 3.41; *p* < 0.001), whereas the General Ability Index (GAI) did not differ significantly (t_(63)_ = 1.58; n.s.). The cognitive profile data are elaborated further in Iaia et al.’s study [82]. Considering these populations, the General Ability Index (GAI) might be deemed a more suitable measure of intelligence compared to the Full-Scale Intelligence Quotient (FSIQ). The FSIQ could be negatively impacted by declines in both the Working Memory Index and the Processing Speed Index, which are often impaired in adults with specific learning disorders and other neurodevelopmental conditions (e.g., ADHD [83,84]).

Regarding reading, raw scores were converted to z-scores according to standard reference data. As expected, the controls’ performance was close to zero for both speed and errors in the two reading tasks, indicating marginal deviations from the standardization sample, while dyslexic participants had a mean below 1.7 SD for speed in both reading tasks, and over 3 SDs for errors in the passage reading test. As expected, significant differences were observed in fluency and accuracy across all reading tasks (*p* at least <0.05), except for text comprehension, between dyslexic and control participants. Dyslexic participants showed comparable impairment in the word and nonword reading tasks (errors: t_(63)_ = −1.02; n.s.; speed: t_(63)_ = −1.55; n.s.).

### 2.2. Materials

#### 2.2.1. Intelligence Assessment

All participants were assessed with the Italian version [79] of the fourth edition of the Wechsler Intelligence Scale for Adults (WAIS-IV; [85]). The WAIS-IV allows for the estimation of Full-Scale IQ (FSIQ) from the assessment of four main indices: Verbal Comprehension (VCI), Perceptual Reasoning (PRI), Working Memory (WMI), and Processing Speed (PSI).

In addition to Full-Scale IQ (FSIQ) and the four main indices, the battery also allows for the calculation of the General Ability Index (GAI), derived from the subtests for the Verbal Comprehension Index, the Perceptual Reasoning Index, and the Cognitive Proficiency Index (CPI), which comprises the subtests for Working Memory and Processing Speed. For a detailed description of the structure of the WAIS-IV and the various indices derivable from it, readers may refer to the manual.

#### 2.2.2. Reading Assessment

The participants’ reading level was assessed by a standard reading achievement battery (LSC-SUA, [78]), constituted by a meaningful passage reading test and a single word and nonword reading test. The following is the description of the reading battery.

##### Passage Reading

The test consisted of a text passage (entitled “Floripa”), based on content on which all assessed subjects were likely to have low and similar familiarity. This included words of different frequency and linguistic complexity, sentences of similar length and structure to those typically encountered by an adult, and rare, and presumably unfamiliar, terms that necessarily required the use of sublexical processes (such as Floripa, Florianopolis, Cananvierias, and Lagoa, all names of places). The passage was 593 syllables long. Participants were asked to read the text aloud. Reading speed (number of syllables read/time in sec) and accuracy (number of errors, adjusted for text read) were considered. The normative data were based on 667 university students [78].

##### Word and Nonword Reading

The test consisted of reading lists of single words and nonwords aloud as accurately and quickly as possible. There were four word lists, each consisting of 28 stimuli: (1) short high-frequency words; (2) short low-frequency words; (3) long high-frequency words; and (4) long low-frequency words. There were two lists of nonwords, also consisting of 28 stimuli each. The nonwords were derived from a syllabic permutation of a subset of words from the word lists. They were divided into (1) short nonwords, made up of 2/3 syllables, and (2) long nonwords, made up of 4 syllables. Speed and accuracy parameters were considered for both words and nonwords. Speed was assessed in syllables per second (for words: the sum in seconds of partial times obtained from the four lists divided by the 352 syllables making up the word test; for nonwords: the sum in seconds of partial times obtained from the two lists divided by the 176 syllables making up the nonword test). For the accuracy parameter, one error was awarded for each omitted or incorrectly read word or nonword. The normative data were based on 667 university students for the words and 666 for the nonwords [78].

##### Text Comprehension

Participants had to silently read one passage and answer 14 questions related to the text. They were given unlimited time to complete the task, were assured that time was not considered in any way, and were allowed to consult the text.

The task exactly followed the standard procedure used by all Italian standardized reading comprehension tasks, i.e., focusing on the student’s ability to find appropriate information in the text to answer a series of comprehension requests to study comprehension independently from the contributions of decoding and memory of the text.

The test was scored by awarding one point for each correct response.

The normative data were based on 562 students [78].

#### 2.2.3. Spelling Assessment

##### Passage Dictation Test

The Passage Dictation Test comprised a meaningful passage consisting of 7 sentences, 102 words, and 221 syllables taken from the battery for the assessment of SLDs and other disorders in university students and adults (LSC-SUA; [78]). The words varied in frequency and linguistic complexity, including words that required the use of lexical processes (e.g., words containing the phonemic group [kw], which in Italian may be transcribed by orthographic sequences QU, CU, or CQU). For example, [kwɔjo] (leather) is spelled CUOIO, and not QUOIO (a nonlexical spelling error), and [akkwjeʃˈʃɛntsa] (aquiescence), is spelled ACQUIESCENZA, and not ACUIESCENZA (another nonlexical error). The passage was dictated individually by the examiner with a neutral tone, modulating the dictation pace according to the participant’s writing speed (so that they could write everything), and pausing where indicated. Participants were permitted to write in either capital or lowercase letters. No feedback was provided on the accuracy of the written response. Self-corrections were accepted, and final responses were counted.

The manual’s instruction is to assign one point for every incorrect word [78]. The normative data were based on 667 university students. Moreover, an analysis of the types of errors, based on previous studies [65,66,86], was tentatively conducted, with the aim of identifying the nature of the spelling errors. The errors were classified as follows:

Lexical errors: Errors that arise from impaired spellings due to the lexical spelling procedure and consequent over-application of the phoneme-to-grapheme routine (spelling errors that can be pronounced to sound like target words). Belonging to this category are the following types of misspellings:-Phonologically plausible errors (PP): Misspellings that sound like the target words (e.g., ACUIESCENZA instead of ACQUIESCENZA (aquiescence))-Word separation and blending: Violation of graphic units of words (e.g., PER TANTO instead of PERTANTO (therefore). The error classification by Angelelli et al. [65,66,86], originally developed for the scoring of the spelling-to-dictation of single words and nonwords, does not include this type of error. However, errors of word fusion and word separation are typical errors resulting from the overuse of the phonological spelling procedure (see also [87,88]), which is why they have become part of the phonologically plausible error category.

Phonological errors: Errors causing a change in the phonemic makeup of a word reflecting difficulties in phonemic segmentation, phoneme-to-grapheme encoding, or a phonological/graphemic buffer disorder. The following types of misspellings fall into this category:-Errors based on minimal distance features (MD): Substitutions of consonants or vowels with other consonants or vowels that differ in only one single distinctive feature. Doubling of a single consonant or de-doubling of a doubled consonant is also considered in this category.-Context-sensitive sound-to-spelling errors (CS): Errors in the application of context-sensitive sound-to-spelling rules (e.g., SCEMA instead of SCHEMA (scheme)).-Simple conversion errors (SC): Non-minimal distance substitutions (e.g., OSTAPOLO instead of OSTACOLO (obstacle)), omission (e.g., CONTATARE instead of CONSTATARE (to account)), insertions (e.g., CUOGIO instead of CUOIO (leather)), or letter transpositions (INTENRAZIONALI instead of INTERNAZIONALI (international)).

Other errors such as word omissions, word substitutions/insertions, and errors in the application of written conventions (e.g., the use of capital vs. lower-case letters, or methods of heading) were also codified but not entered in the rate of lexical vs. phonological errors. These errors may furnish a general idea of written competence and also adherence to the delivery of the task.

##### Single-Word Dictation Test

In this test, the examiner dictates 8 lists of words in two different experimental conditions: 4 lists in the normal condition (NC) and 4 in the articulatory suppression condition (ASC). The articulatory suppression condition allows the evaluation of the level of automation of the spelling process and the subject’s ability to spell under interference conditions.

Each list consists of 14 words, which vary in length and frequency, obtained from the Corpus and Lexicon of Frequency of Written Italian (ColFIS; [89]). The words are characterized as follows:

Each list is composed of 14 stimuli that vary in length and frequency of use:(1)High-frequency short words, composed of 2 syllables, for a total of 28 syllables for both experimental conditions;(2)Low-frequency short words, composed of 2 syllables, for a total of 28 syllables for both experimental conditions;(3)High-frequency long words, composed of 4/5 syllables, for a total of 61 syllables for both experimental conditions;(4)Low-frequency long words, composed of 4/5 syllables, for a total of 61 syllables for both experimental conditions.

In the NC, the examiner dictates the words aloud at a constant pace, about 2 s per word, but is flexible based on the participant’s writing speed. In the articulatory suppression condition (ASC), the examiner dictates the words at a constant pace of about 3 s per word, and simultaneously the participant has to repeat the syllable “LA” aloud throughout the task.

The participant’s performance is measured based on the number of errors, with one point assigned for each misspelled or omitted word [78]. The normative data were based on 677 university students for the NC and 667 for the ASC [78].

## 3. Data Analysis and Results of Spelling Tasks

Table 2 shows the raw number and the transformed z-scores (based on reference data by [78]) of total errors in the meaningful passages and of single words (in the NC and ASC) for participants with dyslexia and the control participants. The raw error rates of the various typologies are also reported. No z-score transformation was possible since the encoding of error types was experimental and not present in the spelling battery. Below is a detailed description of the analysis and results.

### 3.1. Spelling Performance: Passage Dictation Test

The inspection of total error z-scores showed that, in the meaningful passage task, dyslexic participants were slightly (1 SD) above the mean score of the normative sample, showing a mild deficit, while the controls’ performance was close to zero, indicating marginal deviations from the standardization sample. The t-test conducted on total error z-scores revealed that dyslexic participants significantly underperformed with respect to controls (*p* < 0.001).

The exploration of error type revealed a significantly higher number of lexical errors in dyslexic participants (*p* < 0.05), while no significant differences emerged between groups in terms of phonological errors.

Regarding lexical errors, both phonological plausible errors and word blending/separation were significantly higher in dyslexic participants (*p* at least <0.05). Among the phonological errors, simple conversion errors, although few, were the most common type across both groups. However, there was no significant difference between the groups regarding this type of error. Minimal distance substitutions and context-sensitive errors were predominantly observed only in dyslexic participants (*p* < 0.001).

For minor errors, no significant differences emerged between the two groups, indicating that participants with dyslexia did not omit more words with respect to the control participants, nor introduced new (not dictated) words in the passage. Both the latter indices denote a good adherence to the task in participants with dyslexia. Moreover, they showed an adequate knowledge of written conventions.

### 3.2. Single-Word Dictation

An inspection of the total error z-scores for words dictated in the NC and ASC revealed that dyslexic participants in the ASC performed nearly 2 SDs above the mean score of the normative sample on long words and above 1.80 SDs on short words, indicating a moderate-to-severe impairment in this condition. In contrast, their spelling performance in the NC was only slightly above (1 SD) the normative mean score. The controls’ performance was close to zero in both spelling conditions and for all word types, indicating marginal deviations from the standardization sample. A repeated-measures ANOVA was conducted on the z-scores of the errors, with Group (dyslexic vs. control participants) as the between-subject factor, and three repeated factors: Condition (NC vs. ASC), Stimulus length (short vs. long), and Stimulus word frequency (high vs. low). The analysis revealed a significant main effect of Group, indicating that dyslexic participants generally underperformed compared to the controls (F_(1,63)_ = 45.7; *p* < 0.001) across both conditions.

There was also a significant interaction between Group and Condition (F_(1,63)_ = 25.78, *p* < 0.001), with the articulatory suppression condition affecting dyslexics’ performance, but not that of the control participants. The analysis also indicated a main effect of Length, with more errors on long words with respect to short words (F_(1,63)_ = 13.281, *p* < 0.001), and also a Length by Group interaction (F_(1,63)_ = 19.27; *p* < 0.001), indicating that the detrimental effect of length was present only in the dyslexics’ performance (difference in accuracy between long and short words (Δ) −0.82; *p* < 0.001), while control participants had comparable accuracy in spelling short vs. long words (Δ = 0.07; n.s.).

The Condition by Length by Group interaction was also significant (F_(1,63)_ = 3.70, *p* < 0.05; see Figure 1). Post hoc analysis demonstrated that articulatory suppression worsened dyslexics’ performance, but not that of the control group, regardless of short or long stimuli (*p* < 0.001 with respect to controls in both conditions). The detrimental effect was particularly evident for short stimuli, since in the NC dyslexic participants were as accurate as the controls (mean difference, Δ = 0.17; n.s.), while in the ASC their errors increased significantly with respect to the control participants (Δ = 1.63; *p* < 0.001), indicating that the articulatory suppression condition also interfered in the spelling of short stimuli. Regarding long words, dyslexics were more error-prone with respect to the controls both in the NC and the ASC (Δ = 1.78 and 2.62, respectively; both *p* < 0.001) However, their performance on long words further deteriorated under the ASC compared to the NC (NC vs. ASC Δ = 1.01; *p* < 0.01).

Furthermore, the analyses revealed a significant interaction between Condition, Word Frequency, and Group (see Figure 2; F_(1,63)_ = 7.65; *p* = 0.01). Post hoc comparisons indicated that the ASC worsened dyslexics’ performance for both stimuli (at least *p* < 0.001), with a larger effect for high-frequency words with respect to low-frequency ones (Δ = 2.37 for high-frequency words and Δ = 1.67 for low-frequency ones). Control participants had only marginal and non-significant variations in performance passing from the NC to the ASC for both high- and low-frequency words (Δ = 0.12 and 0.19, respectively). In the NC, dyslexic participants performed only slightly but not significantly worse than the controls for both high- and low-frequency words (Δ = 0.64 and Δ = 0.64, respectively; *p* = 0.05 and *p* = 0.06, respectively).

## 4. Discussion

This study aimed to examine the persistence of spelling deficits in highly educated adults with dyslexia in Italy across two different spelling-to-dictation tasks (meaningful text and single words varying in lexical and sublexical complexity), and by introducing an experimental condition, the ASC, to interfere with their naturally developed sublexical compensatory strategy.

The results clearly indicated that in spelling a meaningful passage, dyslexic participants underperformed compared to the controls, despite the comparison with the normative reference data showing only mild difficulties. Their misspellings were orthographic in nature, with a prevalence of lexical errors compared to phonological ones. Among the latter, simple sound-to-spelling conversion errors and minimal distance errors, though few, were still higher than those displayed by control participants, which were almost null. The low number of phonological errors is of interest, considering that the probability of making phonological errors is higher than lexical errors. Hypothetically, a spelling error can occur during the conversion of each phoneme into a grapheme, whereas a lexical misspelling may occur generally once per word at most.

Moreover, the findings clearly demonstrated that when spelling isolated words under normal conditions, dyslexic participants performed within the reference norms and as accurately as control participants across all stimuli (short words, high- and low-frequency words), except for long words, where their spelling difficulties were evident. The introduction of the articulatory suppression condition significantly compromised dyslexics’ spelling performance, indicating a moderate-to-severe impairment across stimuli compared to the reference normative data. Comparisons with control participants revealed that the articulatory suppression significantly impaired dyslexics’ performance on short stimuli, reducing the usual sublexical advantage associated with them, and exacerbated misspellings on long words, which are usually more error-prone in the sublexical spelling procedure. Moreover, articulatory suppression disproportionately affected dyslexics’ performance on high-frequency words compared to the controls, indicating a reduction in the typical advantage associated with these words (due to the ability to rely on both lexical and sublexical procedures for spelling).

Overall, the analysis of both spelling tasks suggests that, in adults with dyslexia who are highly educated and dealing with a transparent orthography, such as Italian, only mild spelling deficits are evident under normal spelling conditions. This holds true when spelling a passage, where the semantic context might offer support, as well as when spelling single words, where such support is not available. In languages like Italian, sublexical spelling can ensure high levels of accuracy, particularly with regular stimuli, even among children with dyslexia [60,65]. However, deficiencies in the automatization of phoneme-to-grapheme encoding and the retrieval of orthographic representation persist and become apparent when phonological analyses (and the maintenance of the resulting phonological specifications) are interfered with.

The present results align with and extend previous findings [8], showing that articulatory suppression conditions accentuate the difficulties experienced by dyslexic individuals, resulting in slower and less accurate performances compared to the controls in the interfering condition. In addition, in our study, clear effects of word length and word frequency emerged, with a greater impact of the interfering situation on the spelling of short stimuli and high-frequency ones. For these stimuli, the articulatory suppression condition significantly reduced the accuracy advantage they have under normal spelling conditions.

This study may provide insights into the mechanisms underlying spelling difficulties in dyslexic individuals and potential areas of weakness. In the investigation by Colombo et al. [74], which aimed to explore the impact of different dual tasks and the potential role of disrupted phonological processes on the spelling of normal adults, poorer spelling performance was observed under the articulatory suppression condition compared to the simultaneous tapping task or normal conditions. The spelling impairment manifested as longer response times, a more significant length effect, and variations in error patterns. According to the authors’ analysis of error type and location within words, a characteristic distribution was revealed, indicating a deficit at the level of the graphemic buffer. These findings align with those of computational studies employing the Competitive Queuing model, as detailed in Glasspool and Houghton [90]. Generally, it is believed that the grapheme buffer stores the final spelling. However, if the deficit lies at the level of the grapheme buffer, the result is a disorder related to the assembly and selection of letter sequences, while the upper and lower levels of the orthographic process remain apparently intact. In such cases, errors occurring in words and nonwords appear similar, with little or no influence from word frequency or grammatical class. The findings of our study do not entirely support the hypothesis that articulatory suppression can directly interfere with the level of the grapheme buffer [74]. Articulatory suppression significantly worsened dyslexics’ performance on short words and high-frequency words compared to the controls (i.e., simpler stimuli for highly educated adults). It also exacerbated misspellings of long words. These findings suggest that articulatory suppression diminishes the usual sublexical advantage associated with short stimuli and increases the difficulty in processing long words. Furthermore, it reduces the typical lexical advantage associated with high-frequency words. In a study on Japanese children [76], who were writing using both ideographic and regular sound-to-spelling syllabic scripts, it was found that articulatory suppression had no apparent effect on ideographic writing. However, it did impair writing in the syllabic script, which requires phonological segmentation and phonological-to-orthographic encoding. Interestingly, in the study on acquired spelling disorders by Folk and Jones [77], articulatory suppression led to a reduction in phonologically plausible misspellings and in the phoneme-to-grapheme probability effect. This latter effect refers to better spelling accuracy on words containing segments with a high probability of phoneme-to-grapheme mapping compared to those with low probability. Both outcomes indicated the successful disruption of the sublexical spelling process, since it is assumed that phonologically plausible misspellings and FG probability effect arise within the sublexical procedure. Moreover, the authors observed an increase in lexical substitution errors, providing evidence that when the contribution of the sublexical system is disrupted, there is an increase in lexical competition, at least in already weakened lexical procedures.

Indeed, according to the “simultaneous activation” hypothesis [57], there exists an interaction between the two spelling procedures involved in a given spelling stimulus. Cooperative interactions occur when both routines agree on spelling, as in the case of regular word spelling (as used in the present study). Both the lexical and sublexical spelling procedures activate candidate graphemic units from a shared pool of elements in the graphemic buffer. Moreover, lexical information can boost the activation of a sublexical item, providing an advantage, for example, in spelling high-frequency words. For instance, in situations where multiple semantically related orthographic lexemes are activated (such as LEOPARD, LION, TIGER, and JAGUAR), the sublexical activation of the graphemes L-E-P-E-R-D biases the activation in favor of the correct lexeme, LEOPARD.

However, a few notes of caution are also needed. Further studies are necessary for the administration of words varied for regularity of transcription (unpredictable transcription words vs. regular words), as well as nonwords under both normal and interfering conditions. A very recent study by some of the authors [91], examined the spelling of words with regular transcription and unpredictable transcription words containing typical or atypical segments. The findings confirmed a lexical deficit in adults with dyslexia, who exhibited greater sensitivity to the frequency and regularity of transcription. Moreover, in the articulatory suppression condition, adults with dyslexia showed the greatest difficulties, with unpredictable transcription with atypical segments (the ones relying solely on the lexical spelling procedure) compared to those with typical segments and regular words. The results suggest reduced lexical processing skills and an overreliance on the sublexical procedure in Italian adults suffering from developmental dyslexia. A limitation of the present study is that it did not explore the efficiency of the lexical reading procedure or characterize reading deficit using ad hoc experimental paradigms (e.g., vocal reaction times) to highlight the parallelism between the spelling and reading processes. Additionally, caution is needed since the study participants were university students, i.e., highly educated and functioning adults. It is uncertain whether less educated and competent adults would demonstrate difficulties in spelling under dictation in simple conditions.

Apart from the theoretical speculations, the present results also need some clinical and educational considerations. Firstly, they confirm the clinical usefulness of using tasks that involve interference conditions such as articulatory suppression, at least in compensated adults dealing with a transparent orthography. These tasks assist in identifying persistent deficits in spelling acquisition, complementing the diagnostic process that relies heavily on the medical and educational history of adults. They enable the identification of current performance issues, enhancing the diagnostic accuracy and understanding of individuals’ spelling abilities.

Furthermore, from an educational perspective, the results highlight the importance of considering extremely challenging dual-task situations in these individuals, especially when processes have partial overlaps. In light of this, providing the necessary time to perform them sequentially or considering other possible compensatory tools, rather than penalizing the presence of misspellings, is important.

Overall, the present study enriches the evidence on the persistence of spelling problems in adulthood, even among highly educated individuals dealing with a transparent orthography, providing insights into the possible mechanisms underlying their long-lasting difficulties.

## Figures and Tables

**Figure 1 brainsci-14-00712-f001:**
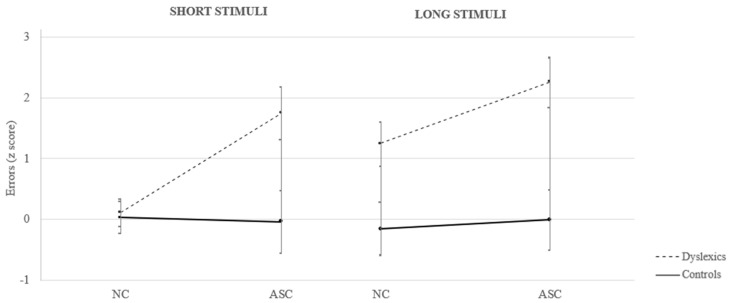
Performance of dyslexic and control participants on short vs long stimuli in Normal Condition (NC) and Articulatory Suppression Condition (ASC). Bars represent the standard deviations (SDs).

**Figure 2 brainsci-14-00712-f002:**
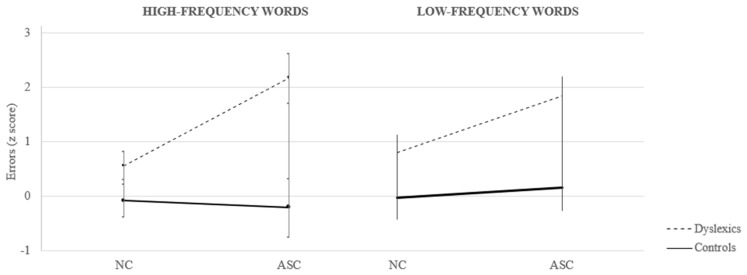
Performance of dyslexic and control participants on high- vs low-frequency words in Normal Condition (NC) and Articulatory Suppression Condition (ASC). Bars represent the standard deviations (SDs).

**Table 1 brainsci-14-00712-t001:** Differences between students with Dyslexia and those with typical development (control group) in age, cognitive profile, and reading tasks. FSIQ = Full Scale Intelligence Quotient; GAI = General Ability Index.

	Dyslexic Participants		Control Participants			
	M (SD)		M(SD)		t (63)	*p*
Age	22.2 (3.07)		22.8 (2.43)		0.85	0.4
FISQ (WAIS-IV)	99 (12.65)		110 (11.67)		3.41	<0.001
GAI (WAIS-IV)	104 (12.71)		109 (11.08)		1.58	0.12
**Reading tests**	**Raw scores**	**z scores**	**Raw scores**	**z scores**		
Passage (syllable/s)	4.20 (0.58)	−1.70 (0.57)	5.99 (0.54)	0.04 (0.53)	12.55	<0.001
Passage (errors)	7.95 (3.63)	3.72 (2.19)	2.17 (1.05)	0.24 (0.63)	−8.01	<0.001
Single words (syllable/s)	2.96 (0.69)	−1.78 (0.67)	4.40 (0.81)	−0.51 (1.12)	5.68	<0.001
Single words (errors)	3.84 (3.00)	0.71 (1.33)	1.11 (1.34)	−0.50 (0.60)	−4.42	<0.001
Single nonwords (syllable/s)	1.79 (0.41)	−1.12 (2.58)	2.92 (0.49)	−0.06 (0.64)	2.09	<0.05
Single nonwords (errors)	5.68 (4.02)	0.96 (1.51)	2.37 (2.02)	−0.29 (0.76)	−3.94	<0.001
Passage comprehension (correct responses)	9.63 (2.36)	−0.22 (0.94)	10.19 (2.09)	−0.00 (0.83)	0.97	0.33

**Table 2 brainsci-14-00712-t002:** Mean errors (raw and z-scores) on the spelling tests for dyslexic and control participants. NC = Normal Condition; ASC = Articulatory Suppression Condition.

	Dyslexic Participants		Control Participants			
	M (SD)		M (SD)		t (63)	*p*
	**Raw scores**	**z-scores**	**Raw scores**	**z-scores**		
Meaningful Passage	7.26 (3.3)	1.33 (1.1)	3.37 (1.4)	−0.02(0.5)	13.71	<0.001
Error type						
Lexical errors	5.37 (2.8)		2.59 (1.4)		7.05	<0.05
Phonological errors	3.32 (2.7)		1.15 (1.4)		3.81	0.06
Subtype of Lexical errors						
Phonological Plausible	4.24 (1.9)		2.26 (1.2)		4.56	<0.05
Word blending/separation	1.13 (1.2)		0.33 (0.5)		14.69	<0.001
Subtype of Phonological errors						
Minimal Distance	0.74 (0.9)		0.07 (0.3)		22.03	<0.001
Context-sensitive	0.18 (0.6)		0.00 (0.00)		13.01	<0.001
Simple Conversion	2.39 (2.4)		1.07 (1.3)		2.72	0.10
Minor errors						
Word omission/substitution	0.58 (1.4)		0.21 (0.6)		−0.6	0.55
Written conventions	0.11 (0.3)		0.11 (0.2)		−0.1	0.94
Single Words NC						
HF Short	0.13 (0.3)	0.12(1.1)	0.07 (0.3)	0.01(0.9)		
LF Short	0.47 (0.9)	0.09(1.2)	0.44 (0.6)	0.06(0.9)		
AF Long	1.13 (1.2)	1.0(1.4)	0.19 (0.4)	−0.17(0.5)		
LF Long	2.79 (2.2)	1.5(1.8)	0.78 (0.9)	−0.12(0.6)		
Single word ASC						
HF Short	1.71 (1.6)	1.81(2.3)	0.19 (0.4)	−0.3(0.5)		
LF Short	3.37 (2.3)	1.67(1.8)	1.56 (1.0)	0.22(0.8)		
AF Long	6.03 (3.9)	2.52(2.1)	1.22 (1.5)	−0.1(0.8)		
LF Long	7.29 (3.2)	1.99(1.2)	2.33 (2.0)	0.1 (0.9)		

## Data Availability

The data are unavailable due to privacy.

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
