# Peer review of "The Enduring Challenge of Literacy Issues in Adulthood: Investigating Spelling Deficits among Dyslexic Italian University Students"

_brainsci, 2024, doi:10.3390/brainsci14070712_

Round 1

Reviewer 1 Report

Comments and Suggestions for Authors

Dear author,

The study deals with an interesting and little-studied issue: the distribution of writing errors in individuals with reading disorders. To improve the quality of the manuscript, revision is needed.

1. When selecting subjects by reading performance, the inclusion criterion you chose is the 15th percentile. However, this casts doubt on the validity of categorizing individuals with reading skill levels between the 10th and 15th percentiles as dyslexic. Rather, these are nonspecific reading disorders. Justify the validity of expanding the range of dyslexia.

2. It is difficult to compare reading speed with other authors' data because you measure speed in syllables per second rather than words per minute, which is the more traditional way of measuring reading speed. It would be worth adding the equivalent of speed in more traditional units of measurement.

3. Descriptive statistics on the results of spelling tasks should be tabulated.

Author Response

Sincerely, 

Francesca Vizzi

Reviewer 2 Report

Comments and Suggestions for Authors

The authors examined the spelling performance of adult Italian dyslexics with two tasks: passage dictation and word spelling. In the latter, participants spelled short vs. long and low- vs. high-frequency words, both under normal and articulatory suppression conditions. Dyslexics underperformed controls in both tasks. Articulatory suppression exacerbated difficulties in short and in high-frequency words.

The study has results that might interest some readership. However, the motives to run this particular study as well as the meaning of the results seem to need clarification, this including streamlining, rewriting and reorganization of ideas. Please note that I am not concerned with the quality of language – just the reasoning.

1.        The goal is not clear:

-According to the authors, the goal is “to explore the spelling profile of Italian university students…” (lns 194-195). I am not sure whether this qualifies as a research goal, given that it can hardly be missed: some exploration will always exist, even if the impact/contribution is null.

-A few lines ahead, the authors mention a more specific interest, which is to “specify the possible loci of difficulties” (198, also 208).  This could be a goal, but then my question is whether the Authors succeeded in doing that (what are the loci?). In lns 590-596, the Authors mention that articulatory suppression may interfere with the retrieval of lexical representations while in lns 499-500 they refer to “reduced lexical processing capacity with over-reliance on the sublexical route”. The question is: why should the interference on lexical processing be a problem, if sublexical processing is the dominant mechanism?

-It is suggested throughout the introduction that little is known adult dyslexia in Italian (eg, ln 148). Is this also part of the goal? If it is, I would suggest the authors explain why this is important. If it is because of the transparency of the orthography, then I would expect to see comparisons with other languages.

2.        In line with (1), I do not see any reference to the hypothesis in the introduction (the loci are…Italian dyslexics, unlike others, are characterized by... ). It would be important to have one. In case the study is purely exploratory and has no hypotheses, then state that. However, in that case, I wonder why the literature on dyslexia in Italian readers and/or transparent orthographies is presented in the introduction.

3.        The justification for the design (independent variables like word length, frequency or articulatory suppression) is distributed across the introduction and the discussion (lns 481-489, 505-524, 553-589). Presenting this in the discussion means presenting it too late. It was particularly striking to me that the proper justification for manipulating articulatory suppression was postponed to the discussion (553-589). Therefore, I would suggest that the authors justify all manipulations before the methods.

4.        Depending on the authors’ response to (1), and going beyond the simple statement of results (this is clear clear), what are the contributions of this study?

Minor

-I don’t think we have two studies here. We have two tasks within one study.

-the word “score” in the YY axis is misleading, since we are talking about errors.

-In table 1, I guess the authors mean “raw” and not “row”?

Comments on the Quality of English Language

No relevant issues detected

Author Response

Sincerely,

Francesca Vizzi

Reviewer 3 Report

Comments and Suggestions for Authors

Review brainsciences Dyslexia

 The Enduring Challenge of Literacy Issues in Adulthood: investigating Spelling Deficits Among Dyslexic Italian University 3 Students

#Francesca Vizzi, #Marika Iaia, Maria Diletta Carlino , Chiara Valeria Marinelli,  Marco Turi and Paola Angelelli  

The study examines spelling deficits among compensated adult Italian speaking dyslexic students. Italian has a transparent orthography. For this, they tested Thirty-eight Italian dyslexic university students  and matched controls,  They did WAIS tests, , then a  first study, : Passage Dictation Test; and a second study with single words : normal condition (NC) and an articulatory suppression condition (ASC).

Results showed: in  Study 1 dyslexic underperformed compared, exhibiting significantly more errors, particularly phonologically plausible; in study 2 Articulatory suppression worsened dyslexics' performance also in  short stimuli, (while comparable accuracy to controls under normal conditions)

The study is well and classically done. It has the privilege to focus on spelling It does not offer very new data but confirm other studies. The most interesting result is for me effects of word length and word frequency on spelling performances. The authors show that dyslexia impair specially performances in  the interfering situation on the spelling of short stimuli and high-frequency ones, suggesting also some non-automatized processes at the lexical and sublexical level.  I have some comments,  mentioned under

Introduction

is clear, the background is well presented and the influence of transparence of orthography is well presented. the reason why to choose spelling task is mentioned. I do however find that the reason for choosing the paradigms (Spelling, different word length articulatory suppression) lack of clarity, as well as the theoretical hypotheses which drove the study. I would find nice to drive the theoretical reasons other than “We tried this paradigm.”

Method is precise ad well described.

Just a point: If I remember well, there are different type of dyslexia.  Can the authors give one or two sentence concerning this point on dyslexic participants?

Results well presented and sound

Discussion:

I like the articulation of the discussion and their point on simultaneous activation theory.  Two remarks

Please give a comment on the WAIS results (since they have been done)

And I wonder if these results are such because of the population is severely dyslexic.  Since Italian is a very transparent language, I can imagine that only a minority of initially dyslexic patients, remained impaired. This could be discussed

Author Response

Sincerely,

Francesca Vizzi

Reviewer 4 Report

Comments and Suggestions for Authors

This is a very interesting work, which uses a linguistic criterion to analyze errors in written language.

But, it's a hard job to read, it's confusing. It needs to be simplified and more coherent.

Methodologically, if the objective of the research is to identify the differences between Italian university students with dyslexia and those with normal reading levels, it requires:

1) Review the criteria for inclusion in both groups. The procedure followed

2) Specify the dependent variables to be analyzed. They should refer to the description of spelling errors.

3) Justify the analysis procedure.

Authors should assess what is necessary to respond to the research objective. The rest may just lead to confusion.

There are doubts about the variables used and the analysis procedures. The consequence is that it is difficult to assume that the results serve the purpose of research.

The discussion should refer to the research objective, for this, they should discuss the results on spelling errors.

Authors need to review the assessments they make in the discussion. These assessments may not be supported by the data they present.

Author Response

Sincerely,

Francesca Vizzi

Round 2

Reviewer 2 Report

Comments and Suggestions for Authors

The authors addressed all my comments and made important changes to improve the quality of the manuscript. 

Comments on the Quality of English Language

no minor issues detected.

Reviewer 4 Report

Comments and Suggestions for Authors

The authors have made an important modification of the initial work. I think the article has improved and can be published.

It has minor problems. Especially when it comes to references. As an example, it can be observed that the instructions for references are not taken into account.

There are major errors in references (e.g., the year of publication or citing a book chapter as if it were a journal article). Some examples are attached. All references should be thoroughly reviewed.